# Polarization Gradient Effect of Negative Capacitance LTFET

**DOI:** 10.3390/mi13030344

**Published:** 2022-02-22

**Authors:** Hao Zhang, Shupeng Chen, Hongxia Liu, Shulong Wang, Dong Wang, Xiaoyang Fan, Chen Chong, Chenyu Yin, Tianzhi Gao

**Affiliations:** Key Laboratory for Wide-Band Gap Semiconductor Materials and Devices of Education, School of Microelectronics, Xidian University, Xi’an 710071, China; zh3298846083@outlook.com (H.Z.); slwang@xidian.edu.cn (S.W.); crotar@163.com (D.W.); fxyxdu@163.com (X.F.); 18829029042@163.com (C.C.); yin_chenyu@163.com (C.Y.); gaotianzhisoso@163.com (T.G.)

**Keywords:** NC-LTFET, subthreshold swing, ferroelectric gate oxide, polarization gradient

## Abstract

In this paper, an L-shaped tunneling field effect transistor (LTFET) with ferroelectric gate oxide layer (Si: HfO_2_) is proposed. The electric characteristic of NC-LTFET is analyzed using Synopsys Sentaurus TCAD. Compared with the conventional LTFET, a steeper subthreshold swing (SS = 18.4 mV/dec) of NC-LTFET is obtained by the mechanism of line tunneling at low gate voltage instead of diagonal tunneling, which is caused by the non-uniform voltage across the gate oxide layer. In addition, we report the polarization gradient effect in a negative capacitance TFET for the first time. It is noted that the polarization gradient effect should not be ignored in TFET. When the polarization gradient parameter g grows larger, the dominant tunneling mechanism that affects the SS is the diagonal tunneling. The on-state current (I_on_) and SS of NC-LTFET become worse.

## 1. Introduction

Due to the limitation of SS (60 mV/dec) at room temperature, detrimental effects, such as short channel effect, higher off-state current (I_off_) and subthreshold swing, would appear in the continuous miniaturization of complementary MOS technology [1,2]. It is no longer feasible to reduce power consumption by lowering the supply voltage [3,4]. As a breakthrough, the TFET with a gate-controlled reverse biased p-i-n diode structure is one of candidates for the next generation of low-power devices because of its lower I_off_ and steep subthreshold slope [5,6,7,8]. Accordingly, it is a meaningful work to study TFET. Till now, some structures with overlapping gate/channel/source are proposed to increase the I_on_ and improve the subthreshold characteristic by increasing the effective tunneling area. Others use hetero-materials such as Si/Ge_x_Si_1−x_ to form a shorter tunneling path to achieve steeper SS and greater I_on_ [9,10,11,12]. However, all the above methods are based on the band-to-band tunneling mechanism to change the n factor of SS to obtain better subthreshold characteristics. The expression of SS is as follows [13]:(1)$$SS=\frac{\partial V_{GS}}{\partial \log_{10}I_{DS}}=\underset{m}{\underset{︸}{\frac{\partial V_{GS}}{\partial \psi_{s}}}}\times \underset{n}{\underset{︸}{\frac{\partial \psi_{s}}{\partial \log_{10}I_{DS}}}}=(1+\frac{C_{d}}{C_{ox}})\times \frac{kT}{q}\ln10$$
where $\Psi_{S}$ is the channel surface potential, C_d_ is the channel capacitance and C_ox_ is the gate oxide capacitance.

Recently, ferroelectric materials were introduced into the device as a gate oxide layer to obtain a body factor m < 1 and extend the steep-slope region, which provides a new idea for the design and optimization of TFET [13,14,15]. Hu et al. [16] reported a negative capacitance vertical-tunnel FET based on GaAs_0.51_Sb_0.49_/In_0.53_Ga_0.47_As with large on-current. Saeidi et al. [17] presented a ferroelectric planner TFET with a minimal subthreshold swing. A novel silicon-based dual source U-shaped channel TFET with negative capacitance (NCDU-TFET) was proposed by Wang et al. [2] However, these studies only consider the effects of remnant polarization, coercive electric field and ferroelectric layer thickness on device performance. In the actual ferroelectric layer, the polarization gradient effect is also very important, it should not be ignored. Although Kao et al. [18] have investigated the influence of polarization gradient effect on Fin-FET, the related work about TFET is hardly reported.

LTFET with the ferroelectric gate oxide layer, namely, NC-LTFET, is proposed in this article. Compared with the traditional LTFET, NC-LTFET has a steeper SS (18.4 mV/dec) and greater I_on_ (2.4 × 10^−7^ A/μm). We then study the working mechanism of the NC-LTFET. Unlike traditional LTFET, NC-LTFET with a small polarization gradient parameter will be turned on by line tunneling at low voltage instead of diagonal tunneling. Finally, the electrical characteristic parameters of NC-LTFET with different g values are analyzed. This will be of great help in understanding the working mechanism of negative capacitance recessed gate tunneling field effect transistors.

## 2. Materials and Methods

Figure 1 shows the schematic of LTFET and the proposed NC-LTFET structure. To improve the performance of LTFET, Si: HfO_2_ (SiO_2_ doped hafnium oxide) is regarded as gate oxide in the proposed structure. The manufacturing process of Si: HfO_2_ thin film is compatible with the existing CMOS technology, which lays a good foundation for the preparation of high-performance negative capacitance tunneling transistors. Kim et al. gives the key process steps for manufacturing LTFET [9]. Meanwhile, the manufacturing process of the ferroelectric materials (Si: HfO_2_) could be obtained from reference [19]. It needs to convert the process steps into a process of ferroelectric material growth when depositing the gate oxide layer. Device parameters of LTFET and NC-LTFET are listed below: gate length, L_g_, is 40 nm; gate oxide thickness, T_ox_, is 2 nm; N^+^ pocket thickness, T_p_, is 5 nm; drain region height, H_D_, is 20 nm; source region height, H_S_, is 40 nm; and the buried oxide height, H_B_, is 80 nm. The doping concentrations of source, drain, pocket and channel are 1 × 10^20^ cm^−3^, 1 × 10^18^ cm^−3^, 5 × 10^18^ cm^−3^ and 1 × 10^15^ cm^−3^, respectively. The gate work function is 4.43 eV and drain voltage is 0.5 V.

The simulation of the NC-LTFET is carried out on a Sentaurus simulator. To calculate the band-to-band tunneling and ferroelectric polarization effect, the non-local band-to-band tunneling (BTBT) model and Landau–Khalatnikov equation are applied in this simulation. Compared with the applicable scope of local model, non-local model was usually described as the spatial variation of the energy bands and it is appropriate for simulating arbitrary tunneling barriers involving nonuniform electric field and abrupt/graded heterojunctions. To obtain the performance of all Si-based TFET devices correctly, this paper uses the approach proposed by Biswas et al. [20] with high accuracy. The fitting coefficients of calibrated model are A_path_ = 1.63 × 10^14^ cm^−3^s^−1^, B_path_ = 1.47 × 10^6^ Vcm^−^^1^ and the corresponding reduced mass is m_r_ = 0.033 × m_o_.

The Landau–Kalashnikov equation and Gibb’s free energy U of ferroelectric materials/system are described as follows:(2)$$-\rho \frac{dP}{dt}=\nabla_{P}U$$
(3)$$U=\alpha P^{2}+\beta P^{4}+\gamma P^{6}-EP-g{\left|\nabla P\right|}^{2}$$
where ρ is the kinetic coefficient associated with polarization-switching dynamics. Chatterjee et al. [21] reports that the intrinsic delay of a doped hafnium oxide-based ferroelectric is negligible in digital circuits. Thus, the kinetic coefficient of the ferroelectric system is zero. P is the total polarization, E is the electric field of ferroelectric material, and α, β and γ are static coefficients for a ferroelectric material. g is a coupling coefficient for the polarization gradient term of the free energy, typical value of g ranges from 10^−6^ to 10^−2^ cm^3^/F [22]. Figure 2a shows RC network to calibrate the ferroelectric parameters, which consists of an external resistor (capacitor) and a ferroelectric capacitor. According to the recently published experimental data of ferroelectric materials Si: HfO_2_, the calibration of the P–E curve is shown in Figure 2b [19]. The calibration parameters for the ferroelectric material Si: HfO_2_ are $\alpha =-4.32\times 10^{10}\mathrm{cm}/F$, $\beta =3.86\times 10^{19}{\mathrm{cm}}^{5}/{\mathrm{FC}}^{2}$, $\gamma =6.48\times 10^{29}{\mathrm{cm}}^{9}/{\mathrm{FC}}^{4}$ and the corresponding $E_{C}=1\;\mathrm{MV}/\mathrm{cm}$ and $P_{r}=11.2\;\mu C/{\mathrm{cm}}^{2}$.

By combining Equations (2) and (3), the relationship between the electric field and polarization is obtained, as shown in Equation (4):
(4)$$E=2\alpha P+4\beta P^{3}+6\gamma P^{5}-2g\Delta P$$

Here, ∆ is the Laplace operator. Considering that the gate charge is equal to the polarization near the interface between gate and gate oxide $P\equiv Q_{g}$, the voltage across the ferroelectric materials could be expressed as below [18]:(5)$$\begin{array}{ll} V_{FE} & =T_{ox}\times (2\alpha Q_{g}+4\beta Q_{g}{}^{3}+6\gamma Q_{g}{}^{5}\\ & -2g\frac{d^{2}Q_{g}}{d^{2}x}-2g\frac{d^{2}Q_{g}}{d^{2}y}) \end{array}$$
where T_ox_ is the thickness of ferroelectric oxide. Moreover, mobility with doping and electric field dependence, Shockley–Read–Hall recombination, bandgap narrowing and Fermi statistics are also considered. A detailed description of these device parameters can be found in Table 1, and they are used in the following sections unless stated otherwise.

## 3. Results

### 3.1. Performance and Mechanism Comparison of NC-LTFET with LTFET

Figure 3a shows the transfer characteristic curve of NC-LTFET and LTFET at V_DS_ = 0.5 V. In order to intuitively display the change of transfer characteristic curve, the NC-LTFET transfer characteristic curve moves 0.1 V to the right, as the illustration of Figure 3a shows. The SS of NC-LTFET and LTFET is 18.3 mV/dec and 33 mV/dec at the range of drain current from 4 × 10^−17^ to 1 × 10^−9^ A/μm, respectively (the SS calculated in this manuscript is average subthreshold swing). The much smaller SS shows that NC-LTFET is more sensitive to the influence of low gate voltage. Moreover, because of its large I_on_ (2.4 × 10^−7^ A/μm at V_GS_ = 1 V), NC-LTFET would have a stronger driving capability in digital circuits. Meanwhile, the smaller I_off_ (4 × 10^−17^ A/μm at V_GS_ = 0 V) value, the smaller the static power consumption in the off state. Therefore, NC-LTFET is one of the most promising low-power devices, and further research is needed.

To clarify the working mechanism of NC-LTFET, Figure 3b exhibits the electrostatic potential along A-B direction when V_g_ = 1 V. Figure 3c gives the band diagram of NC-LTFET and LTFET along A-B direction. Unlike the traditional high-K gate oxide materials (HfO_2_), ferroelectric material (Si: HfO_2_) has large remnant polarization, and high depolarization electric field is throughout the ferroelectric bulk. The high depolarization electric field would induce the increase of surface potential in device, which can be clearly observed in Figure 3b. However, the variation can not only cause the downward bending of the energy band, but also expand the range of the overlap area between the conduction band and the valence band, as shown in Figure 3c. The electron/hole transmission probability, T, can be calculated through the Wentzel–Kramers–Brillouin (WKB) method, which is given by the expression in (6) [23]:
(6)$$T\approx \exp(-\frac{4\lambda \sqrt{2m^{*}}\sqrt{E_{g}{}^{3}}}{3q\hbar (E_{g}+\Delta \phi )})$$
where m* is the carriers’ effective mass, E_g_ is the bandwidth and λ is the effective tunneling length. LTFET and NC-LTFET have the overlapping regions $∆\emptyset_{1}$ and $∆\emptyset_{2}$, respectively. It is easy to infer that the NC-LTFET with greater energy overlapping region $∆\emptyset_{2}$ will produce a larger band-to-band tunneling rate in the pocket area. Figure 3d reveals the electron band-to-band tunneling rate of NC-LTFET is larger than that of LTFET along the C-D direction, which confirms the previous speculation. Finally, NC-LTFET with larger current density is shown in Figure 3e.

Figure 4a shows the electric field distribution of the LTFET near the source region at V_GS_ = 0.4 V and V_DS_ = 0.5 V. Next, the electrostatic potential of NC-LTFET in Figure 4b is extracted along the cutline C-D marked in the inset. Additionally, the electrostatic potential of the LTFET is extracted in the same way. Obviously, the crowd electric field at the corner of the source region would elevate the surface potential near the corner, which is given as black squares in Figure 4b. However, contrary to the case of LTFET, NC-LTFET has a greater electrostatic potential near the pocket region, instead of the corner region. The change is attributed to the inconsistent rise of the surface potential, which is caused by the difference in the voltage amplification effect along the C-D direction. It could be observed that the surface potential near the pocket region of NC-LTFET is 0.2 V higher than that of traditional LTFET at V_GS_ = 0.4 V and V_DS_ = 0.5 V, but the value is 0.03 V far away from pocket region. At the same time, according to the previous discussion, the greater the surface potential, the greater the downward bending of the energy band, the greater the energy overlap region, and the greater the probability of electron tunneling. Therefore, the introduction of ferroelectric materials would significantly enhance the line tunneling. Figure 4c represents the significant enhancement of line tunneling in NC-LTFET when the gate voltage changes from 0.25 to 0.4 V. Compared with the LTFET, NC-LTFET will be the first to be turned on by line tunneling as shown in Figure 4c. Additionally, the tunneling generation rate of line tunneling in NC-LTFET is much larger than that of diagonal tunneling in LTFET at V_GS_ = 0.25 V. Ultimately, the transfer characteristic curve with ultra-steep SS would be obtained under the low gate voltage.

Figure 5a not only shows the change of gate potential along the C-D direction, but also exhibits the change of surface potential under the gate oxide and the change of the voltage V_ox_ across the gate oxide layer. As previously described, the surface potential of LTFET near the corner region will increase because of the electric field crowding effect. For this reason, the voltage across the gate oxide of LTFET will be the smallest near the source corner and the largest near the source region. Like LTFET, the voltage assigned to the gate oxide layer near the source region of NC-LTFET is higher than voltage assigned to the gate oxide layer near the source corner. Hence, the ferroelectric gate oxide layer near the pocket region would generate larger polarized charges under applied voltage, and establish higher electric field. It can be concluded that the ferroelectric polarization at the corner of the recessed gate would be much smaller than that of near the source region as the black arrow shown in Figure 5b. Finally, as the red arrow shows in Figure 5c, NC-LTFET has a strong voltage amplification effect in the pocket area and starts to weaken at the corner when the gate voltage is 0.4 V. With the help of the non-uniform V_Fe_, the performance of the device would be optimized and the working mechanism would be changed.

### 3.2. The Impact of Polarization Gradient on NC-LTFET Performance

To further investigate the influence of polarization distribution on the electrical characteristic of NC-LTFET, different polarization gradient parameter g is selected as the variable, and static ferroelectric parameters still remained ($\alpha =-4.32\times 10^{10}\mathrm{cm}/F$, $\beta =3.86\times 10^{19}{\mathrm{cm}}^{5}/{\mathrm{FC}}^{2}$, $\gamma =6.48\times 10^{29}{\mathrm{cm}}^{9}/{\mathrm{FC}}^{4}$). Note that instead of multidomain state, sections of the FE material are stable in the case of different g values, and the single domain with inhomogeneous polarization (polarization gradient) is considered in this simulation. These g values are 10^−4^ cm^3^/F, 5 × 10^−3^ cm^3^/F, 10^−2^ cm^3^/F, respectively. Figure 6a shows that g has a great impact on the transfer characteristic curve of NC-LTFET, especially in the subthreshold region. When g value changes from 10^−4^ to 10^−2^ cm^3^/F, the SS and drain current would be deteriorated. SS increases with the increase of g value, and the drain current decreases with the increase of g. Therefore, the smaller the g value, the greater the improvement in the device performance. It should be noted that, when the gate voltage is 0.5 V, the drain current is saturated and it decreases a little. Figure 6b shows the SSs with different g values. These SSs are 18.4 mV/dec, 20.8 mV/dec and 23.2 mV/dec, respectively. Additionally, they are calculated by fixing the current range from 4 × 10^−17^ to 1 × 10^−9^ A/μm.

Figure 7a shows the ferroelectric polarization of NC-LTFET along C-D direction at different g values. The method of extracting data has been described above. Figure 7b,c show the electrostatic potential and the conduction band energy along C-D direction at different g values, respectively. As shown in Figure 7a, with the increase of the polarization gradient parameter g, the polarization near the pocket region decreases, and the polarization near the corner of NC-LTFET increases. As we discussed previously, the degree of ferroelectric polarization determines the value of surface potential. Accordingly, the electrostatic potential near corner regions increases and near pocket regions decreases with the increase of the polarization gradient parameter g at V_GS_ = 0.25 V. Hence, as shown in Figure 7c, the conduction band energy near the corner is bound to reduce and the conduction band energy near the pocket region is bound to increase. As shown in Figure 7c,d, the tunneling will open at a location with a lower conduction band energy along the C-D direction. When the polarization gradient parameter g grows larger, the dominant tunneling mechanism that affects the SS will change from line tunneling to the diagonal tunneling. Both line tunneling and diagonal tunneling appear in the position along C-D direction. The tunneling mechanism dominated by diagonal tunneling will deteriorate the device performance.

## 4. Conclusions

This article proposes an L-shaped gate tunneling field effect transistor with a ferroelectric gate oxide layer. Firstly, the characteristics of traditional LTFET and negative capacitance LTFET are compared and discussed. According to the curves obtained in A-B and C-D directions, it can be seen that NC-LTFET has higher potential in the pocket area and deeper band bending will occur, which can help NC-LTFET have a larger current and a steeper SS. In addition, by studying the relationship between voltage and potential at the corner and near the pocket region, it is found that NC-LTFET has strong voltage amplification effect near pocket region. This leads NC-LTFET turns on at first by line tunneling at low gate voltage. Finally, different polarization gradient parameters are used to study the potential and conduction band energy of NC-LTFET. The experiment shows that the higher the parameter g, the smaller the line tunneling rate that will be generated, and the role of diagonal tunneling will be greater, which will further worsen I_on_ and SS. 

## Figures and Tables

**Figure 1 micromachines-13-00344-f001:**
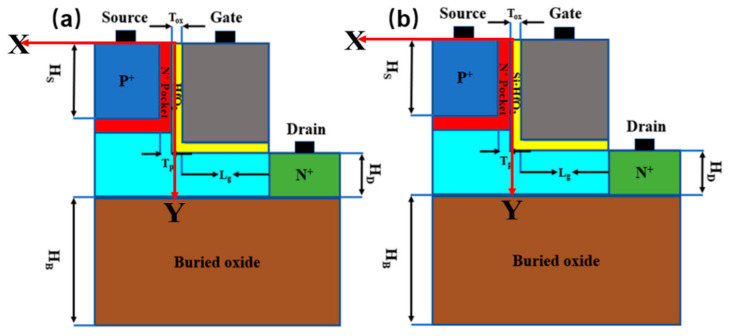
(**a**) The cross-section view of LTFET and (**b**) NC-LTFET.

**Figure 2 micromachines-13-00344-f002:**
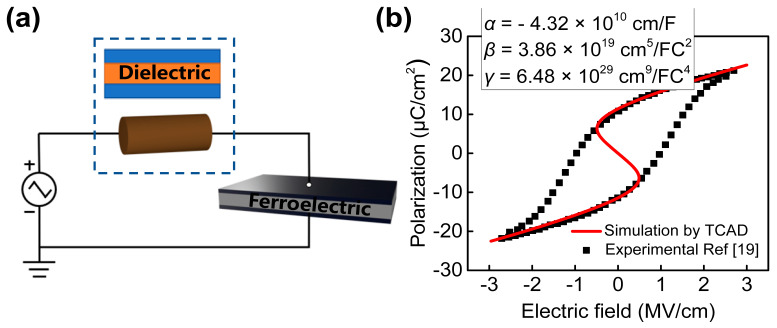
(**a**) Negative capacitance measurement circuit, (**b**) comparison between simulated P–E curve with experimental results for Si: HfO_2_ [19].

**Figure 3 micromachines-13-00344-f003:**
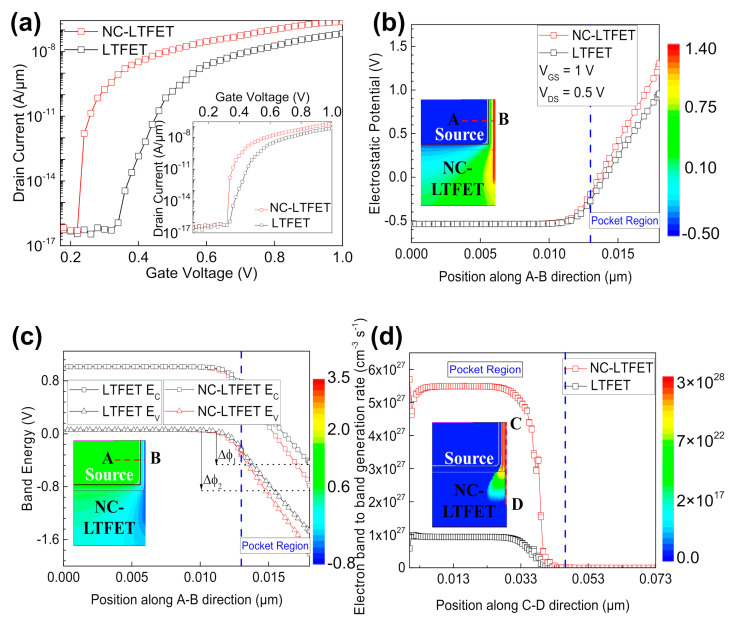
(**a**) The transfer characteristic curve of NC-LTFET and LTFET, (**b**) the electrostatic potential of NC-LTFET and LTFET along A-B direction, (**c**) the band energy of NC-LTFET and LTFET along A-B direction, (**d**) the band-to-band generation rate of NC-LTFET and LTFET along C-D direction, (**e**) total current density distribution of NC-LTFET and LTFET.

**Figure 4 micromachines-13-00344-f004:**
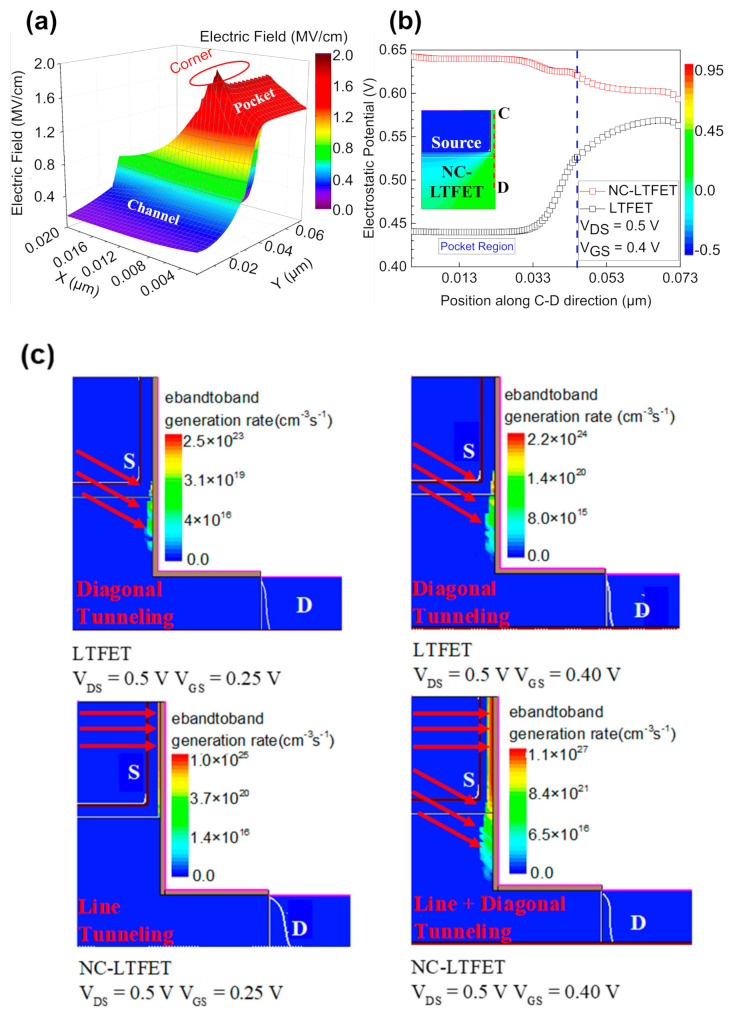
(**a**) The electric field distribution near the source region of LTFET at V_GS_ = 0.4 V and V_DS_ = 0.5 V, (**b**) the surface potential under the gate oxide along the C-D direction at V_GS_ = 0.4 V and V_DS_ = 0.5 V and (**c**) the electron band-to-band generation rate of LTFET and NC-LTFET at low gate voltage.

**Figure 5 micromachines-13-00344-f005:**
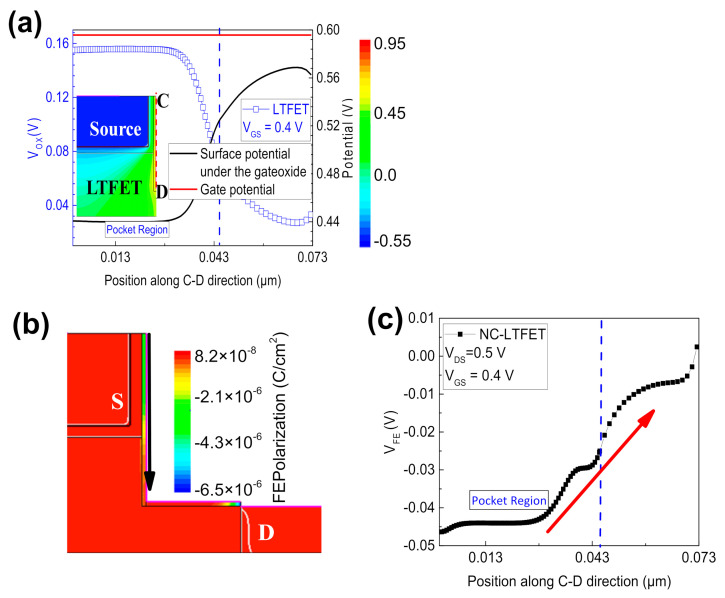
(**a**) The surface potential under the LTFET’s gate oxide along the C-D direction, the gate potential of LTFET and the voltage across gate oxide layer of LTFET are also considered, (**b**) polarization distribution of NC-LTFET at V_GS_ = 0.4 V and V_DS_ = 0.5 V and (**c**) the voltage across the ferroelectric gate oxide layer along the C-D direction.

**Figure 6 micromachines-13-00344-f006:**
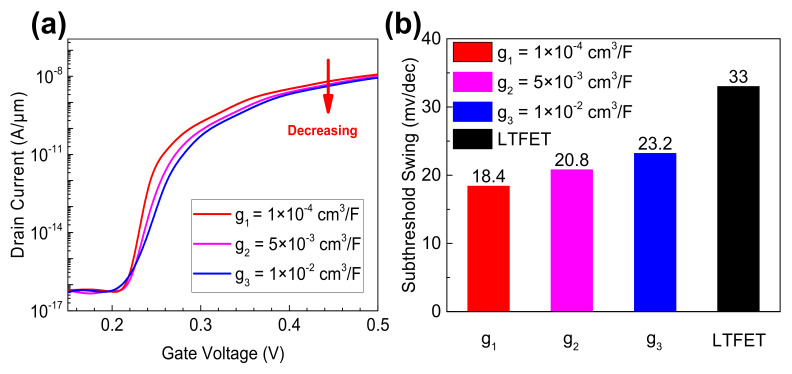
(**a**) The transfer characteristic curves under different g values and (**b**) the influence of g on SS.

**Figure 7 micromachines-13-00344-f007:**
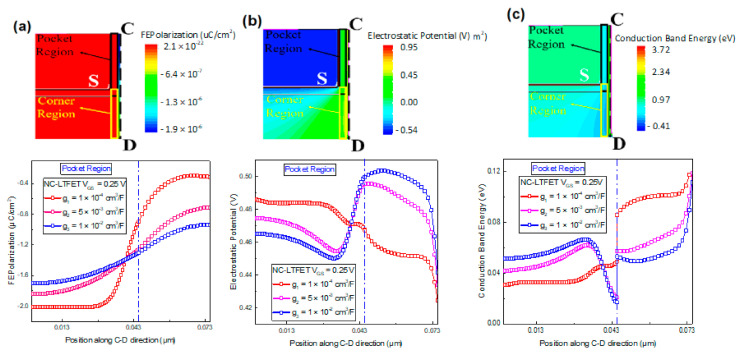
(**a**) Ferroelectric polarization of NC-LTFET along C-D direction at V_GS_ = 0.25 V, (**b**) the electrostatic potential of NC-LTFET along C-D direction, (**c**) the conduction band energy of NC-LTFET along C-D direction and (**d**) the tunneling position of NC-LTFET with different g values at V_GS_ = 0.25 V and V_DS_ = 0.5 V.

**Table 1 micromachines-13-00344-t001:** Device parameters of LTFET and NC-LTFET.

	LTFET	NC-LTFET
Gate length (L_g_)	40 nm	40 nm
Source height (H_S_)	40 nm	40 nm
Drain height (H_D_)	20 nm	20 nm
Gate oxide thickness (T_OX_)	2 nm	2 nm
N+ pocket thickness (T_p_)	5 nm	5 nm
Buried oxide thickness (H_B_)	80 nm	80 nm
Source doping (N_AS_)	10^20^ cm^−3^	10^20^ cm^−3^
Drain doping (N_DD_)	10^18^ cm^−3^	10^18^ cm^−3^
Channel doping (N_AC_)	10^15^ cm^−3^	10^15^ cm^−3^
Pocket doping (N_DP_)	5 × 10^18^ cm^−3^	5 × 10^18^ cm^−3^
Coercive field (E_c_)		1 MV/cm
Remnant polarization (P_r_)		$11.2\;\mu C/{\mathrm{cm}}^{2}$
$\mathrm{Gate}\;\mathrm{oxide}\;\mathrm{dielectric}\;\mathrm{constant}\;(\varepsilon_{r})$	22	26

## Data Availability

Data are contained within the article.

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
