# Peer review of "Polarization Gradient Effect of Negative Capacitance LTFET"

_micromachines, 2022, doi:10.3390/mi13030344_

Round 1

Reviewer 1 Report

After reviewing your assigned manuscript entitled with “Polarization Gradient Effect of Negative Capacitance LTFET”, I will give the following questions to the authors for revisions:

  1. Page 1, Introduction, line 33, Eqn. (1) should be revised as SS = ∂VGS ∂(log 10IDS) = ∂VGS ∂ψS × ∂ψS ∂(log 10IDS) = (1 + CS/Cins ) × kT/ q ln 10 = m × n, and explain the symbols of ψS, CS, Cins, respectively, which factors control the parameters m and n.
  2. Page 3, line 90, Eqn. (2):r , please use italic font for all letters.
  3. Page 6, line 166-169: Please explain in detail from the Fig. 4 (c) or references why the introduction of ferroelectric materials would significantly enhance the line tunneling and the area of line tunneling is much larger than that of diagonal tunneling.
  4. Page 6, line 187-188: “the ferroelectric polarization at the corner of the recessed gate would be much smaller than that of near the source region as shown in Fig. 5(b). As shown in Fig. 5(c), NC-LTFET has a strong voltage amplification effect in the pocket area and starts to weaken at the corner when gate voltage is 0.4V.” Please indicate the descriptions on those graphics.
  5. Page 7, line 199: The title of 3.2 is the same with that of 3.1. Please check again and revise the title of the section 3.2.
  6. Page 8, line 217~220: “Fig. 7(a) shows the ferroelectric polarization of NC-LTFET along C-D direction at different g values….” Please add the device’s diagram with the position along C-D direction.
  7. Page 8, line 224:”…near corner regions increases and near pocket regions…” In order to understand tendencies of NC-LTFET along C-D direction with different g values, please indicate the corner and pocket regions on the device’s diagram.
  8. Page 9, line 235-238: “When the polarization gradient parameter g grows larger, … from line tunneling to the diagonal tunneling. Both line tunneling and diagonal tunneling appeared in the position along C-D direction.” It’s better to add marks of line tunneling and diagonal tunneling on the diagrams of Fig. 7 (d) for explaining clearly this paragraph.

Author Response

We thank you very much for giving us an opportunity to revise our manuscript, we appreciate to revise our manuscript, we appreciate editor and reviewers very much for their positive and constructive comments and suggestions on our manuscript entitled “Polarization Gradient Effect of Negative Capacitance LTFET” (ID: micromachines-1591110). And I will answer your questions one by one below.

Reviewer 2 Report

This manuscript proposes an L-type tunneling field effect transistor with a ferroelectric gate oxide layer. Firstly, the characteristics of traditional LTFET and negative capacitance LTFET are compared and discussed, and the corresponding equations of electric field E and voltage VFE are listed. Then, the electrical characteristics of LTFET and NC-LTFET are studied experimentally. According to the curves obtained in A-B and C-D directions, it can be seen that NC-LTFET has higher potential in the pocket area and deeper band bending will occur. NC-LTFET will produce higher electron band generation rate through line tunneling. In addition, by studying the relationship between voltage and potential at the corner and near the source region, it is found that NC-LTFET has strong voltage amplification effect. Finally, different polarization gradient parameters g are used to study the potential and conduction band energy of NC-LTFET. The experiment shows that the higher the parameter g, the smaller the line tunneling rate will be generated, and the role of diagonal tunneling will be greater, which will further worsen Ion and SS. This manuscript still has the following problems, which need to be further improved, and it is suggested to review after revision.

  1. ‘SSave’ and ‘sub-threshold swing (SS)’ in this manuscript need to be clearly explained and distinguished to avoid misunderstanding.
  2. The equations and serial numbers in the manuscript should be aligned and centered in the article.
  3. It is suggested that pictures should be in the form of Figure instead of Fig.
  4. The variables in the equation and the units of variables should be consistent with those in the manuscript. It is suggested to check carefully and modify them.
  5. More detailed analysis and discussion should be made in Figure 2b, and the reasons for the inconsistency with the literature [19] should be given.
  6. In the second part, the equations of QP and VFE need to give a detailed derivation process to make the manuscript more complete.
  7. The experimental contents of subsections 3.1 and 3.2 are different, and it is suggested to give different titles.
  8. In Figure 3, (b) and (c) give the curve in AB direction. Why is the electron band generation rate in A-B direction not given in (d)?
  9. The in Figure 4 (a) shows the X and Y directions, which are suggested to be marked in Figure 2 for ease of understanding.
  10. The conclusion of the manuscript should be summarized in the order discussed above, so as to make the manuscript more organized so that readers can see the conclusion more intuitively.

Author Response

(The authors gave the same response as above.)

Round 2

Reviewer 1 Report

After reviewing your assigned manuscript entitled with “Polarization Gradient Effect of Negative Capacitance LTFET”, I will give the following comments and yellow marks in  the attachment to the authors of the manuscript No. micromachines-1591110. for revisions:

  1. Page 1, line 33, where Ψ? is channel surface potential, Cd is the channel capacitance and Cox is the gate oxide capacitance. Please use italic font for the symbols of ψS, CS, Cox, respectively.
  2. Page 4, line 117, please add a space between 0.5 and V; Page 6, line 172, add a space between VGS = and 0.25 V; Page 7, line 192, add a space between 0.4 and V; Page 7, line 198, add a space between the number and V; Page 8, line 215, add a space between 4 ×10-17 and to; Page 9, line 237, and remove parentheses of d.
  3. Page 10, References, No.1, 4, 7, and 16 of abbreviated journal name use italic font, please check the right format to follow the journal of micromachines.

Reviewer 2 Report

accept